# The Association between Charolais Cows’ Age at First Calving, Parity, Breeding Seasonality, and Calf Growing Performance

**DOI:** 10.3390/ani13182901

**Published:** 2023-09-13

**Authors:** Birutė Šlyžienė, Indrė Mečionytė, Vytuolis Žilaitis, Eglė Butkienė, Lina Anskienė, Evaldas Šlyžius, Giedrius Palubinskas

**Affiliations:** 1Department of Animal Breeding, Faculty of Animal Science, Lithuanian University of Health Sciences, Tilžes Str. 18, LT-47181 Kaunas, Lithuania; indre.mecionyte@lsmuni.lt (I.M.); eglebutkiene1@gmail.com (E.B.); lina.anskiene@lsmuni.lt (L.A.); evaldas.slyzius@lsmuni.lt (E.Š.); giedrius.palubinskas@lsmuni.lt (G.P.); 2Large Animals Clinic, Faculty of Veterinary Medicine, Lithuanian University of Health Sciences, Tilžes Str. 18, LT-47181 Kaunas, Lithuania; vytuolis.zilaitis@lsmuni.lt

**Keywords:** Charolais, beef cattle, birth weight, daily gain

## Abstract

**Simple Summary:**

Meat producers are trying to increase the efficiency of beef production, with a primary focus on genetic and management strategies for beef cattle breeding. Calf birth weight and growth potential, such as weight at different growth periods, are influenced by a variety of genetic and non-genetic factors. The objective of this study was to evaluate the impact of cows’ calving age and newborn calf body weight on the pre-weaning growth rates of calves under the farming conditions of a northeastern European country. The gender of the calf, cows’ age at first calving, cows’ number of calvings, and the breeding and calving seasons are frequently used indicators in beef cattle farming. According to the influence of selected factors in this study, it can be concluded that the body weight and growth rates of newborn calves are significantly influenced by the gender of the calf, the number of calvings, as well as calving and breeding seasons, should be taken into account in future animal breeding studies.

**Abstract:**

The objective of this study was to evaluate the impact of cows’ calving ages and newborn calf body weight on the pre-weaning growth rates of calves under the farming conditions of a northeastern European country. The females were purebred Charolais cows that calved between 2019 and 2022. A total of 130 calves (bulls = 76; heifers = 54) were observed during the evaluation. The investigation showed that, based on gender, bulls had a 4.28% higher birth weight than heifers (*p* < 0.05). The gender of the calves affected their weight, indicating that newborns bulls were significantly heavier. Calf gender affected calf growing performance. Male calves’ weight gain was higher than that of females in the time before weaning (210 days). The highest gain of bulls was observed from 211 days to 365 days after birth. The average daily weight of male calves during this period was 6.16% (*p* < 0.001) higher compared to the period up to 210 days after birth. Heifers had a higher daily weight in the period up to 210 days after birth, and the daily weight gain was 12.9% (*p* < 0.001) higher in this period compared to the period from 211 to 365 days after birth. We observed a higher weight gain in calves from mothers with a higher number of calvings. Being born from cows with four or more calvings had a significant effect on the weight gain of calves in the period up to 210 days (*p* < 0.05). In the period up to 210 days, the weight gain of calves born from cows with two calvings was the lowest and was 9.79% lower compared to calves born from cows with four and more calvings (*p* < 0.05). The best weights were obtained from calves born in the spring and autumn seasons. Calves born in spring, summer, and autumn differed in weight by 0.37 kg. The calves of cows that were inseminated in the autumn season had the highest gain in the period up to 210 days and also from 210 to 365 days (*p* < 0.05). In our study, significantly higher average birth weight of calves was detected in autumn compared to summer and winter (*p* < 0.05). Also, the highest gain was found from 211 to 365 days after birth in the summer season, but the difference was only 1.47% (*p* < 0.05).

## 1. Introduction

Beef animal husbandry is currently one of the fastest-growing agricultural sectors in the world. This market is witnessing progressive growth due to increased demands for beef, population growth, urbanization, and rising incomes [1,2]. Geopolitical and economic changes have led to an increased consumption of high-quality meat in Europe. The rising demand for food, especially meat, encourages farmers to use new strategies in order to improve the efficiency of beef cattle. One of the most popular beef cattle breeds in the world is Charolais. The beef cattle Charolais was developed in France. This breed has been adapted to grow in different geographical places. Today, Charolais cattle can be found in every beef-cattle-producing country. This breed is the most common in Central Europe and Mexico [1,3]. There are differences in herd management, feed, and climate between countries [1]. Several possible factors, including genetic–environmental interactions and differences between management characteristics and geographical regions, cause different production performances in the Charolais breed [4]. It has also been observed that some meat qualities of Charolais cattle depend on climatic conditions [5]. In this context, meat producers are looking for ways to increase the efficiency of beef production. To solve these problems, the genetics and management strategies of beef cattle are the main goals to focus on.

Effective breeding strategies are one of the determining factors in the economic viability of beef cattle production. Selection for a higher breeding value is an effective way of improving fertility traits [6].

The main characteristics influencing efficient beef production are the growth of calves at slaughter age and effective feed intake. Previous research has established relationships between cow milk yield and calf pre-weaning growth [7]. Therefore, the age of a cow at calving affects the development of its calf through the cow’s milk yield, and milk yield estimation in any month is highly related to the calf’s average daily gain from birth to weaning [8]. 

Calf birth live weight and growth potential, such as weight at different growth periods, are influenced by many genetic factors (such as breed) as well as non-genetic factors (such as the gender of the calf [9,10,11], the age at first calving [6,12], parity [11], as well as the calving [7,11] and breeding seasons [12]).

To monitor calves’ growth, it is appropriate to check their weight after birth. It is observed that new-born calf weight is associated with growth rates from birth to weaning. The weight of a new-born calf is affected by genetic predisposition and the weight of the cow [13]. Farmers attempt to maximize calf weight and the value of meat at weaning. The effect of the gender of calves influences their growing performance and meat quality. The average daily gain, width extremities, and thorax depth were found to be higher in males than in females in a previous study [14]. Some authors state that, in reality, bull calves gain weight faster than steer calves [15]. However, meat from beef cows is more tender and more desirable to consumers. The influence of beef gender is illustrated by the fact that calves after castration (steers) have a slower growth rate and lower feed conversion efficiency compared to non-castrated ones [16]. Under farming conditions, the ratio of male calves to female calves is approximately equal. The influence of the gender of the calf on its weight gain is important for practical evaluation purposes. 

In beef cattle herds, it is recommended to control the seasonality of breeding. A controlled calving season is more profitable for beef cattle farmers than year-round calving, as nutrition can be adjusted according to physiological needs [7,17]. The difficulty of calving can be monitored; calves will be more uniform in weight and age, and cows with reproductive disorders will be easier to identify [17]. Previous research has shown the influence of calving season on calving traits. Grings et al. (2008) estimated that the average daily gain was greater (*p* = 0.003) in the late winter than in the early spring calving season [7]. Hozáková et al. (2019) estimated calving season’s influence on offspring in terms of birth weight (*p* < 0.001) and pre-weaning period (*p* < 0.001) [11]. The differences in the weight of calves (at birth, weaning, and 210 days) were estimated when comparing the calving time (early or late) and maturity stage (young or adult) [12].

The rearing of Charolais beef cattle is increasing across all continents [18]. The main reasons for selecting Charolais cattle are high production rates with high weight gain rates [1] and live body weight traits [18]. Also, the economic value of selecting fertility traits is considerable because poor fertility results in low profitability in beef cattle production [6,19]. Selection criteria refer to the reproductive and production aspects in modern selection programs [11,19]. Hence, the objective of this study was to evaluate the impact of cows’ calving age and newborn calf body weight on the pre-weaning growth rates of calves under the farming conditions of a northeastern European country.

## 2. Materials and Methods

This study was conducted at a beef cattle farm located in Lithuania (WGS 55.3592549, 22.6490364). The females were purebred Charolais cows that calved between 2019 and 2022. During the evaluation, a total of 130 calves (bulls = 76; heifers = 54) were monitored.

During the period from October to May, the cows were housed indoors in pens of 15 to 25 individuals on semi-deep litter. During the winter period, the animals were provided with a feed ration consisting of hay and corn silage in ad libitum amounts. During the grazing period from May to September, the calves were kept on the pasture together with their mothers, as indicated in Table 1. The animals had access to water ad libitum. 

From 2 to 5 months of age, the calves were fed with a grow starter feed in an ad libitum manner. The composition of the grow starter feed typically includes a combination of ingredients to provide a balanced diet for the growing calves; the components of this feed are shown in Table 2. These ingredients (corn, wheat, soybean meal, rapeseed meal, limestone, molasses, sodium chloride, and monocalcium phosphate) are typically combined in specific proportions to create a balanced diet that meets the nutritional requirements of growing calves during the 2-to-5-month age range.

The feeding regimen for the calves between 5 and 12 months involved the use of homemade forage. The feed composition typically includes the following components, shown in Table 3: grains, extracted rapeseed meal, beans, sodium chloride, and fodder chalk. The amounts of feed consumed by the calves varied based on their age. The feeding amounts were as follows: 5–7 months: ad libitum amount, meaning the calves were allowed to eat as much as they desired during this period; 7 months: 2.0 kg/day; 8 months: 2.5 kg/day; 9 months: 3.0 kg/day; 10 months: 3.5 kg/day; and 11–12 months: 4.0 kg/day.

The cows were inseminated with frozen–thawed semen of proven Charolais sires after a waiting period of 60 days. The farmers aimed to improve specific traits when selecting Charolais bulls, including looking at their growth, weight, muscle development, conformation, carcass quality, fertility, reproductive performance, and temperament. Three Charolais bulls were used, each exhibiting specific trait indexes. The details of each bull’s trait indexes are as follows: calving ease: index ranged from 108 to 112; growth potential: index ranged from 102 to 108; weaning proofs: index ranged from 107 to 111.

In this study, the cows (*n* = 52) were divided into groups according to their calving age and number of calvings. Age at first calving was defined as the number of months from a cow’s birth to its first calving. Considering the influence of age at first calving, the cows were divided according to their age at first calving: up to 27 months (*n* = 11), 28–31 months (*n* = 18), 32–35 months (*n* = 13), and over 36 months (*n* = 10). Considering the influence of number of calves, the cows were divided into the following categories: first calving (*n* = 38), second calving (*n* = 35), third calving (*n* = 32), and fourth calving and more (*n* = 25).

To assess the influence of breeding season, we divided the cows according to the period in which they were bred: spring, summer, autumn, and winter.

During the evaluation, the calves were grouped into four groups based on the calving season: spring calves (*n* = 32) born from March to May, summer calves *(n* = 39) born from June to August, autumn calves (*n* = 28) born from September to November, and winter calves (*n* = 31) born from December to February.

The calves were weighed within 24 h after calving (birth weight) and at 210 days and 365 days after birth (based on the Charolais cattle breeding program for the years 2018–2028, http://lmga.lt/naudinga-informacija/veisimo-programos/, accessed on 1 March 2023, Lithuania).

Birth and daily weights were determined using verified and certified scales in a fixed cattle capture pen (Shanghai Youhua Weighing System CO., Ltd., Changhai, China, model OIML XK3190-A12E, Certification no. R76/2006-CN1-09.01). The daily weight variations were determined as a function of the difference between weights divided by the number of days between weightings.

The calculation of calf daily weight gain from birth to 210 days was performed according to the formula:
kg/day = (actual weight (kg) − birth weight (kg)/210 days.

Calf daily weight gain from 211 days up to 365 days was calculated according to the formula:
kg/day = (actual weight (kg) − weight (kg) 210 day)/155 days.

Statistical analysis was performed using the IBM SPSS Statistics software (version 25.0, IBM, Munich, Germany). The normal distributions of the variables were assessed using the Kolmogorov–Smirnov test. One-way analysis of variance (ANOVA) was used to determine statistically significant differences between the means of the groups (age at 1st calving, number of calvings, breeding and calving seasons). Multiple comparisons of group means were calculated using the post hoc Tukey’s test. Differences in the mean values between groups of different genders were analyzed using Student’s *t*-test. The differences were considered statistically significant at *p* < 0.05. A linear model was created to describe the relationship of birth weight and gain of calves with the following fixed effects: gender of the calf, age at first calving, number of calvings, breeding season, and calving seasons. The following formula was used:
Y_ijklm_ = μ + G_i_+ FC_j_+ NC_k_ + BS_l_ + CS_m_ + e_ijklm_
where

Y_ijklm_—dependent variable (gain of calves up to 210 days, gain of calves from 211 to 365 days, birth weight of calves);μ—mean value of dependent variable;G_i_—fixed effect of ith gender of calf;FC_j_—fixed effect of jth first calving age;NC_k_—fixed effect of kth number of calvings;BS_l_—fixed effect of lth breeding season;CS_j_—fixed effect of mth calving season;e_ijklm_—random error.

## 3. Results and Discussion

### 3.1. The Influence of Genderon the Birth Weight of Calves and Weight Gain

The investigation showed that regarding gender (Figure 1), bulls had 4.28% higher birth weight than heifers (*p* < 0.05). The relationship between calf gender and birth weight of calf has also been reported by other researchers [20]. According to our data, the gender of the calves affected their weight, showing that new-born bulls were significantly heavier. While the effect of gender has also been demonstrated by other authors, the mechanism of how gender affects calf birth weight still is not clear [21]. Overall, growth performance is influenced by the gender of the calf. We found that male calves’ weight gain was higher than female calves’ weight gain during the time before weaning (210 days). The highest weight gain of bulls was observed from 211 days to 365 days after birth. The average daily weight of male calves during this period was 6.16% (*p* < 0.001) higher compared to the period up to 210 days after birth. Heifers had a higher daily weight by 12.9% in the period up to 210 days after birth, compared to the period from 211 to 365 days after birth (*p* < 0.001). The improved growing performance of bulls might be associated with the pronounced anabolic effects of androgens. It could be attributed to the fact that beef cattle producers have safely used various types of growth-enhancing technologies to enhance anabolic activity [22]. Also, bull calves have greater plasma insulin than heifer calves. The differences in postnatal change in plasma leptin due to the gender of the calf may affect the appetite center of the hypothalamus, thereby influencing appetite and weight gain [23].

### 3.2. The Influence of Cows’ Age at First Calving on the Birth Weight of Calves and Weight Gain 

The highest average birth weight of calves (Figure 2) was found in the group of cows that calved for the first time until 27 months of age. The average weight of calves born in this group was 2.43% higher (*p* > 0.05) compared to calves born in the group of cows of 28–31 months of age at first calving. The highest average weight of calves was estimated between 211 and 365 days for calves whose mothers calved for the first time when they were over 36 months of age. The average daily weight of these calves during this period was 25.86% higher than that of calves whose mothers calved for the first time between 28 and 31 months of age. Optimizing live weight performance from birth to slaughter is a key objective and is associated with higher profitability [24]. Age at first calving has a negative genetic correlation with the lifetime number of calvings and a positive correlation with calving interval. Correlations with production and type traits are low, except for skeletal development [25]. Increased age of Holstein cows at first calving is associated with a higher calf birth weight [26]. The dependence of calf weight at birth on cows’ age is nonlinear. We found that new-born calf body weight was higher for cows that calved earlier for the first time until 27 months of age, and for cows that calved for the first time when over 36 months of age. The optimal first calving time is influenced by the interaction between genotype and environment [27]. Early calving is associated with dystocia, but late mating of cows predicts economic loss. For the development of optimal calf growth, weight gain is also important. The relationship between cow age at first calving and growth performance of calf is controversial. According to L. Stádník (2008), calves of primiparous cows had significantly the highest body weight but significantly the lowest weight gain [28]. While older cows produced more milk, calves of young, early-calving cows weighed more at 210 days than those born from late-calving cows [12]. In other words, cows’ age at calving did not have an impact on weaning weight, but it affected calves’ weight gain [27]. According to our data, calves born from cows older than 36 months grew faster than those born from younger cows. The lowest growing performance was observed in calves born from cows of 28–35 months of age. The growing performance of calves was weakly associated with newborn calf weight, but unfavorable effects were observed on the weight and growth of calves when heifers calved at 28–35 months of age.

### 3.3. The Influence of Calving Number on Calves’ Birth Weight and Weight Gain

The highest average birth weight of calves was found (Figure 3) in the group with four and more calvings. However, this difference was small at only 1.9%, and was not statistically significant (*p* > 0.05). No statistically significant differences were observed when comparing the average birth weights of calves in the other groups. Cow age or parity was associated with the number of calvings, which was affected by other reproductive characteristics, in particular ‘days open’ [29]. Therefore, the parity of cows is closely correlated with the number of calving, but these two constructs are not the same. According to our results, the number of calvings was positively associated with calf weight. A newborn calf’s weight was associated with survival and dystocia. Calves of intermediate birth weight had a higher survival rate than calves of either low or high birth weights [30]. It is plausible that the cause effect of cows’ age on calf size is related to the maternal environment during the development of the placenta. Calf size at birth and placental weight are positively correlated, but it is still unknown which one controls and signals the growth of the other [21]. The growth of calves and their weaning weight are influenced by several factors, such as genotype as well as other individual factors [31]. We observed the highest weight gain in calves from mothers with a higher number of calvings. Being born to cows in the group with four and more calvings had a significant effect on the gain of calves up to 210 days (*p* < 0.05). The daily gain of calves born from cows with two calvings was the lowest up to 210 days, and gain was 9.79% less than calves born from cows with four and more calvings (*p* < 0.05). During the period from 211 to 365 days, the average weight gain of calves born from cows with four and more calving’s was 11.9% higher than calves born from cows with three and fewer calving’s (*p* > 0.05). The offspring of beef cows with higher parity has a higher possibility of achieving a better growing performance. It is postulated that a newborn calf’s body weight gain is influenced by the mother’s milk yield. Usually, the milk yield of cows is associated with their age at calving. In a previous study, the peak yield in Herefords cows was observed at 8.4 years [32]. The association of milk yield with cows’ age has been noticed in Angus, Charolais, and Hereford cow breeds. It was observed that differences in total milk yield and dry matter due to breed and cow age were highly significant [33]. On the other hand, the body weight of a female cow is linked to their mass index (MI). MI is a measure derived from body mass. It was noticed that calves born from low-MI cows (*p* < 0.05) were heavier at 210-day postpartum than those born from cows with moderate and high MI [1].

### 3.4. The Influence of Breeding Season on Calves’ Birth Weight and Weight Gain 

We found that the best weights were obtained from calves born in spring and autumn (Figure 4). The weight of calves born in winter was 2.97% lower than that of calves born in spring (*p* > 0.05). Calves that were born in spring, summer, and autumn differed in weight by ~0.37 kg. Bitencourt et al. (2020) evaluated the influence of two sub-periods of the calving season (early calving from 06 September to 15 October, and late calving from 16 October to 30 November) and two stages of maturity (young or adult) and found that calves born from early-calving adult cows weighed 4.2 kg, which was higher than the weight of those born from late-calving adult cows [12]. We estimated that cows that were inseminated in the autumn season had calves with the highest gain in the period up to 210 days as well as from 210 to 365 days (*p* < 0.05). Grings et al. (2008) estimated that the average daily gain was higher (*p* = 0.003) in calving occurred in the late winter than in the early spring [7].

### 3.5. The Influence of Calving Season on Calves’ Birth Weight and Weight Gain

In our study, the highest mean birth weight of calves (Figure 5) was detected in autumn compared to summer and winter (*p* < 0.05). In the period up to 210 days, the average gain of calves born in the summer season was 21.42% and 9.72% higher compared to the gain of calves born in the spring and winter seasons, respectively (*p* > 0.05). Also, the highest gain was found from 211 to 365 days after birth in calves born in the summer season, but the difference was only 1.47% (*p* < 0.05). In another study with Charolais cows, the season of calving influenced birth weight (*p* < 0.05) and weights at 120 days and at 210 days (*p* < 0.001) [11]. The results of our study were inconsistent with the findings in the study by Hozakova et al. (2019), in which the heaviest calves were born in spring. A study in Hungary with beef cattle breeds showed that the weaning weight of calves could be affected by factors such as breed, age of cow, season of birth, and gender [34]. Szabo et al. (2006) estimated that calves born in summer reached the highest weight at 205 days after birth, while the lowest weight was observed in calves born in winter. In a study with Charolais cows, Contreras et al. (2015) reported that the highest weaning weight was estimated in September–October (*p* < 0.01) [3].

## 4. Conclusions

An effective breeding strategy is critical to the economic viability of beef cattle production. Animal breeding plays a vital role in the management and productivity of beef cattle herds, which affects the profitability and sustainability of the operation. Thus, beef cattle producers must prioritize these traits in their breeding and herd management strategies. The gender of the calf, cows’ age at first calving, cows’ number of calvings, and breeding and calving seasons are the most frequently used indicators in beef cattle farming. Regarding the relationship between the selected factors in this study, it can be concluded that newborn calves’ body weight and the growth rates of calves are significantly influenced by the gender of the calf, number of calvings, and breeding and calving seasons. Based on the results of our findings, these indicators should be taken into account in future animal breeding studies.

## Figures and Tables

**Figure 1 animals-13-02901-f001:**
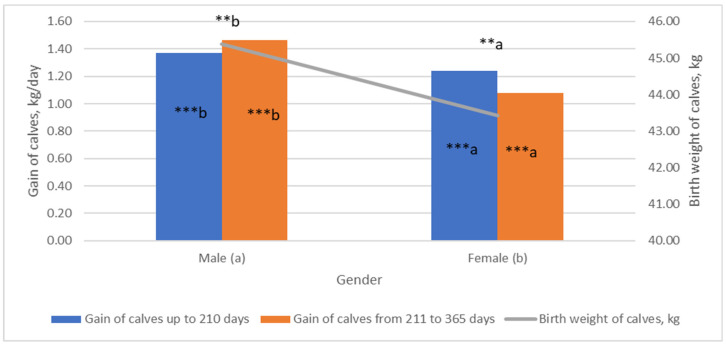
The relationship between calf gender and birth weight and gain. (a, b) Markings with different letters indicate significant mean differences in the investigated indicators between genders (male—(**a**) and female—(**b**)). *** *p* < 0.001; ** *p* < 0.01.

**Figure 2 animals-13-02901-f002:**
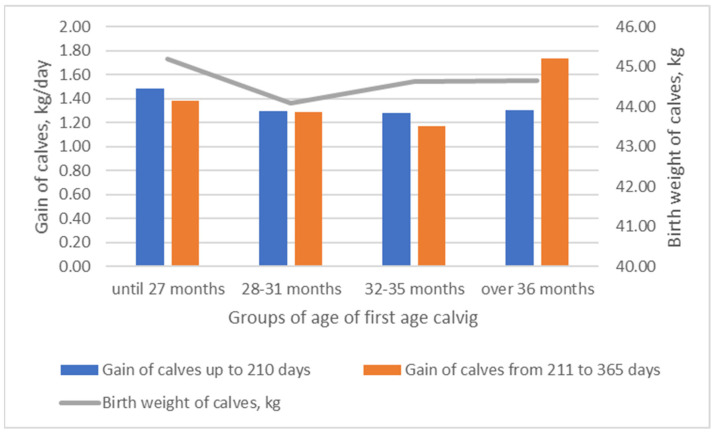
The influence of age of cows at first calving on birth weight and gain of calves.

**Figure 3 animals-13-02901-f003:**
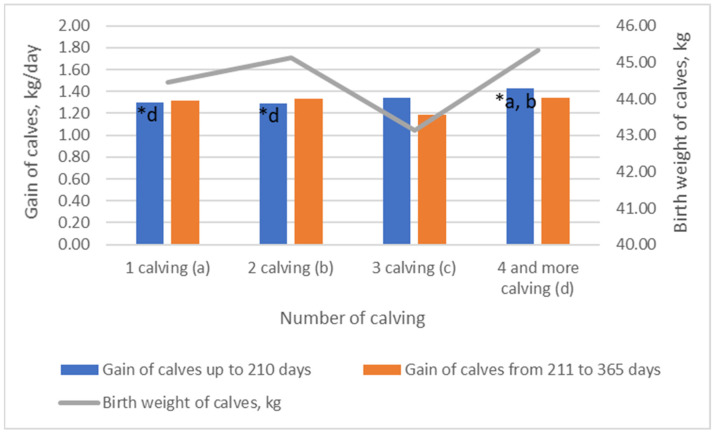
The influence of cows’ number of calvings on birth weight and gain of calves. (a, b, d) Markings with different letters indicate significant mean differences in the investigated indicators between groups (1 calving—(**a**), 2 calvings—(**b**), 3 calvings—(**c**), and 4 and more calvings—(**d**)).; * *p* < 0.05.

**Figure 4 animals-13-02901-f004:**
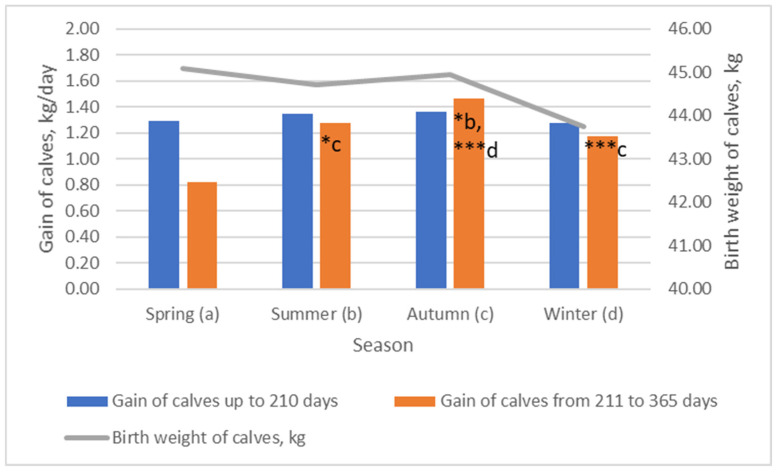
The influence of breeding season on birth weight and gain of calves. (b–d) Markings with different letters indicate significant mean differences in the investigated indicators between groups (spring—(**a**), summer—(**b**), autumn—(**c**), and winter—(**d**)). *** *p* < 0.001; * *p* < 0.05.

**Figure 5 animals-13-02901-f005:**
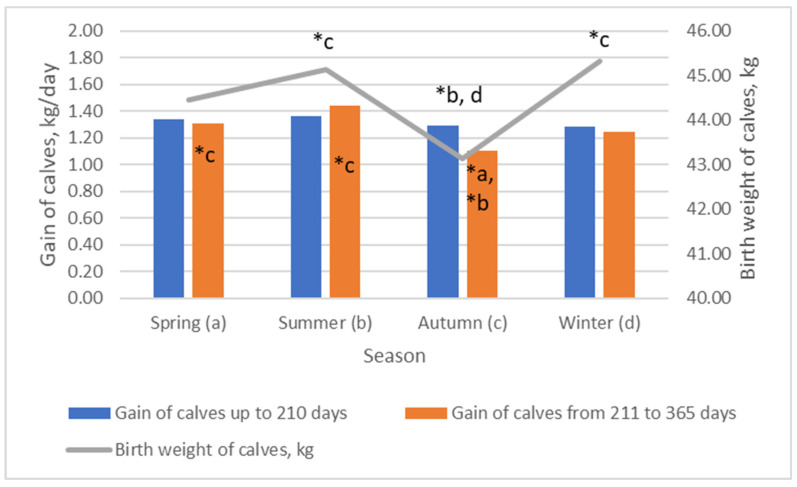
The influence of calving season on birth weight and gain of calves. (a–d) Markings with different letters indicate significant mean differences in the investigated indicators between groups (spring—(**a**), summer—(**b**), autumn—(**c**), and winter—(**d**)).; * *p* < 0.05.

**Table 1 animals-13-02901-t001:** The components of the total mix ratio (TMR). Daily feeding for each cow.

Indicators	Pasteur Grass	Hay	Corn Silage
Dry matter, g/kg	261	841	331
Ash, g/kg	84	59	48
Crude protein, g/kg	118	106	85
Crude fiber, g/kg	247	252	188
Crude fat, g/kg	33	22	28
Total sugar, g/kg	107	134	297
Acid detergent fiber (ADF), g/kg	297	366	204
Acid detergent lignin, g/kg	43	417	218
Neutral detergent fiber (NDF), g/kg	449	493	365
Net energy for lactation (NEL), MJ	6.31	8.49	6.60
Fermentation rate	-	-	28.65

**Table 2 animals-13-02901-t002:** The exact chemical composition of the grow starter feed used in this study.

Indicators	Value
NE, MJ/kg	11.00
Crude protein, %	18.10
Lysine, %	0.90
Crude fiber, %	3.50
Crude fat, %	2.50
Green ash, %	5.00
Calcium (Ca)	1.04
Phosphorus (P)	0.70
Sodium (Well)	0.21
Magnesium (Mg)	0.33
Zinc, mg/kg	105.00
Manganese, g/kg	95.00
Copper, g/kg	25.00
Cobalt, g/kg	1.10
Iodine, g/kg	2.00
Selenium, g/kg	0.30

**Table 3 animals-13-02901-t003:** The chemical composition of the home-made forage.

Indicators	The Homemade Forage
Dry matter, g/kg	891
Crude ash, g/kg SM	120
Crude protein, g/kg SM	186
Crude fiber, g/kg	50
Crude fat, g/kg	18
Total sugar, g/kg	414
Net energy for lactation (NEL), MJ	7.35
Calcium, g/kg SM	18.89
Phosphorus, g/kg SM	6.83
Magnesium, g/kg SM	5.75
Sodium, g/kg SM	12.27
Potassium, g/kg SM	6.20
Chlorine, g/kg SM	1.32
Sulfur, g/kg SM	4.15

## Data Availability

The data presented in this study are available within the article.

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
