# Peer review of "The Association between Charolais Cows’ Age at First Calving, Parity, Breeding Seasonality, and Calf Growing Performance"

_animals, 2023, doi:10.3390/ani13182901_

Round 1

Reviewer 1 Report (Previous Reviewer 1)

The revised manuscript was much improved and has already answered my previous concerns. I don't have any further question and agreed to publish this manuscript.

Author Response

Dear Reviewer,

Thank You for the comments and Suggestions. 

Reviewer 2 Report (Previous Reviewer 2)

Dear Authors,

Your manuscript has been improved after your last review and it might be now acceptable for publication after minor revision. Please, check all the Figures. Verify the existence of significant differences and show all of them properly. Use the letters to show the differences and * for significance level. 

Yours sincerely,

Reviewer.

Author Response

Dear Reviewer,

Thank You for the comments and suggestions. 

Reviewer 3 Report (Previous Reviewer 3)

Dear author,

In the text there are some paragraphs in which the language has to be improved. Try to improve the language in the text of materials and methoda  (especialy in the paragraph of statistical analysis) and in the text concerning t=results and discussion. The previous session was better.

Author Response

Dear Reviewer,

Thank You for the comments and suggestions. 

This manuscript is a resubmission of an earlier submission. The following is a list of the peer review reports and author responses from that submission.

Round 1

Reviewer 1 Report

The results of this manuscript are important for the management of Charolais reproduction, to obtain better development of offspring. However, it is only some data cumularing, no any mechanism study.

Reviewer 2 Report

Dear Authors,

Your manuscript titled "Association between Charolais dam's age at first calving, parity, breeding seasonality and calf growing performance" is addressing an interesting topic of research. It is suitable for publication at Animals after major revision. Please, see below a list of comments/suggestions to be applied by you before accepting it.

L2 Replace 'dam's' by another term more appropriate to define them.

L11 Replace 'dam's' by another term more appropriate to define them. 

L22 Replace 'the higher' by 'higher'.

L23 Replace 'calving's' by 'calvings'.

L29 Replace 'a higher' by 'higher'.

L42 Replace 'Charolais developed' by ''Charolais was developed' and 'was adapted' by 'is adapted'.

L62 Replace 'calf weight associated' by 'calf weight is associated' and 'weaning' by 'weaning.'.

L86 Replace 'was to' by 'was then to' and replace 'dam's' by another term more appropriate to define them. 

L36-L88 Review the Introduction section to show the novelty of your research in comparison with previous work done in relation to that by others.

L93 Replace 'dam's' by another term more appropriate to define them. 

L97 Replace 'dam's' by another term more appropriate to define them. 

L98 Replace 'dam's' by another term more appropriate to define them. 

Table 1: Remove line spacing between rows.

Table 2: Remove line spacing between rows.

Table 3: Remove line spacing between rows.

L123 Replace 'from108 to112' by 'from 108 to 112' and 'to108' by 'to 108'.

L124 Replace 'from 107-111' by 'from 107 to 111'.

L125 Replace 'dam's' by another term more appropriate to define them. 

L126 Replace 'calving's' by 'calvings'.

L127 Replace 'dam's' by another term more appropriate to define them. 

L130 Replace 'dam's' by another term more appropriate to define them.

L132 Replace 'dam's' by another term more appropriate to define them.  

L140 Describe the model used and the standard deviation for the measurements of weight done.

L143 Replace 'ac-cording' by 'a-ccording'.

L145 Replace 'day' by 'days'.

L158-L159 Replace 'also indicates' by 'has also been observed in'.

L164-L165 Replace 'aver-age' by 'ave-rage'.

L168 Replace 'bulls associated' by 'bulls might be associated'.

L169 Replace 'Now,' by 'It could be attributed to the fact that'.

L175-L176 Replace 'dif-ferent' by 'di-fferent'.

L180 Replace 'dam's' by another term more appropriate to define them. 

L182 Replace 'dam's' by another term more appropriate to define them. 

L192 Replace 'dam's' by another term more appropriate to define them. 

L193 Replace 'dam's' by another term more appropriate to define them. 

L196 Replace 'dam's' by another term more appropriate to define them. 

L196-L197 Replace 'man-agement' by 'mana-gement'.

L197 Replace 'dam's' by another term more appropriate to define them. 

L203 Replace 'dam's' by another term more appropriate to define them. 

L204 Replace 'dam's' by another term more appropriate to define them. 

Figure 2: Add letters to show significant differences among treatments.

L210 Replace 'dam's' by another term more appropriate to define them. 

L218 Replace "days open" by 'days open'.

L223 Replace 'dam's' by another term more appropriate to define them. 

L226-L227 Replace 'gen-otype' by 'geno-type', 'higher' by 'highest' and 'moth-ers' by 'mo-thers'.

L228 Replace 'dam's' by another term more appropriate to define them. 

L231 Replace 'dam's' by another term more appropriate to define them. 

L231-L233 Replace 'calving's' by 'calvings'.

L233 Replace 'dam's' by another term more appropriate to define them. 

L236 Replace 'dam's' by another term more appropriate to define them. 

L231 Replace 'Therefore,' by 'Hence,'. 

L237 Replace 'of Herefords' by 'in Herefords'.

L239 Replace 'dam' by another term more appropriate to define them. 

L241 Replace 'It is noticed,' by 'It is noticed'.

L243 Replace 'BMI' by 'MI'.

Figure 3: Replace 'of number' by 'of the number' and 'calving's of' by 'calvings from'.

Figure 4: Add letters to spring.

L266 Replace 'a higher' by 'higher'.

L274 Replace 'the beef' by 'beef'.

L276 Replace 'dam' by another term more appropriate to define them. 

Figure 5: Review all letters.

L289-L289 Review the meaning of this sentence and rewrite it.

L291-L293 Replace 'dam's' by another term more appropriate to define them. 

L285-L304 Review all Conclusion section. Try not to show a summary of results. It is needed to provide real interpretation of data for future research trials. Add some implications of the work done.

L321-L417 Review all references according to Animals' instructions for authors.

  The reserach evaluated the impact of 5 factors: gender, age at first calving, parity, breeding season and calving season on calves' birth weights and weight gains of 130 Charolais calves (bulls=76 and heifers=54) weighed during three times at 24 hours, at 210 days and at 365 days after calving.   The major issues found in the article that need to be addressed by the authors are the following:
  - Review the Introduction section to show the novelty of the research conducted herein in comparison with previous works conducted by others. No new ideas are provided to support this study. Most of the factors under study have been previously investigated by other researchers.   - Review the Material and Methods section to describe in detail the equipment used for all the measurements done to allow its reproductibility and explain which was the main criteria considered for the experimental design. Give detailed information about the statistical analysis performed.   - Review the Results section to show the agreement between the text and Figure's significance.   - Review the Conclusions section. Try not to write a summary of results. It is needed to provide a real interpretation of data for future research trials. Add some implications of the work done.   - Review all references following Animal's instructions to the authors.   The minor issues found in the article that need to be addressed by the authors are the following:   - Review gramatical errors, modify table's settings, check figure's significance and text's readability.

Yours sincerely,

Reviewer.

Reviewer 3 Report

Dear author,

Bellow you will find comments  that you have to follow and to do in  your manuscript 

P2L44: [1, 3].  instead of  [1], [3].

P2L59-60: such as gender of calf [8–10], age at the first calving [11, 12], parity [10] as well as the 59 calving [6, 10] and breeding seasons [11]. instead of  such as gender of calf [8]–[10], age at the first calving [11], [12], parity [10] as well as the 59 calving [6], [10] and breeding seasons [11].

P2L75 : [6, 17]. instead of  [6], [17].

P2L92: were monitored instead of  were observed

P4L123-124: from108-112, growth potential: from 102-108 instead of  from108 to112, growth potential: from 102 to108

P4L132: For assessing the influence of the breeding season the dams we divided according to instead of  Assessing the influence of the breeding season we divided the dams according to

P5L178: The influence of cows' age at first calving instead of  The influence age of cows at first calving

P7L228: a higher number of calving’s instead of  a bigger amount of calving’s

P9L288: on the birth period instead of  on the period from birth

*Also I have to inform the authors that reference No 5 is not referred to the text although existing in thw references